# Large-Scale Illegal Bryophyte Harvesting in Protected Areas of East-Central Europe, Hungary: Conservation Implications

**DOI:** 10.3390/plants14243785

**Published:** 2025-12-12

**Authors:** Péter Szűcs, Sándor Rózsa, Marianna Marschall

**Affiliations:** 1Department of Botany and Plant Physiology, Institute of Biology, Eszterházy Károly Catholic University, Leányka út 12, 3300 Eger, Hungary; szucs.peter@uni-eszterhazy.hu; 2Aggtelek National Park Directorate, Tengerszem Oldal 1, 3759 Aggtelek, Hungary

**Keywords:** illegal harvest, bryophyte collection, bryophytes of forest environments, bryoflora, bryophyte gatherers practice, *Hypnum cupressiforme*, criminal offense, nature conservation site, conservation biology

## Abstract

This study provides the first detailed documentation of large-scale illegal bryophyte harvesting within a European nature conservation site. Forested areas of Northeast Hungary are recurrently affected by such activities, with several cases already resulting in official criminal proceedings. Although commercial bryophyte harvesting is not explicitly prohibited within the European Union, it is indirectly constrained by the conservation framework of the Habitats Directive. Our objective was to assess the conservation biological consequences of removing a substantial volume (296 kg air-dry weight; 8.7 m^3^) of bryophytes. Sixteen bryophyte species, including one liverwort and fifteen mosses, were identified in the confiscated material. Harvesters primarily target *Hypnum cupressiforme*, a moss species favored for decorative wreath production, with demand increasing prior to All Souls’ Day in Hungary. Illegal collectors typically operate in small groups within forest stands proximal to settlements, concentrating their activity on the bark of *Quercus* trees and andesite rock surfaces. Terricolous bryophytes and associated soil substrates were entirely absent from the collected material. Comparative analysis revealed that the bryophyte flora of the affected forest stands is more diverse than that represented in the harvested samples. This indiscriminate illegal bryophyte harvest threatens protected forest habitats and necessitates improved monitoring and stricter enforcement.

## 1. Introduction

The miniature size, negligible appearance, challenging taxonomy status, general lack of commercial value, and inconspicuous position in the bryophyte ecosystem have led most people to overlook bryophytes [1,2].

In ethnobotanical aspects, some bryophytes have old stories; we know the most ethnobryological records from China, India, Mexico, and the USA [3,4,5]. Lately, in SE Europe, Bučar and coworkers published the uses and diversity of harvested bryophytes in Croatian traditional nativity scenes in Christmas festivities [6].

Nowadays, bryophytes are widely used for decorative purposes in the floral industry and traditional decorative applications. However, the origin, harvesting conditions, locations, volumes, and species diversity of commercial bryophyte materials are less known or unknown.

Commercial bryophyte harvesting cases were published from the American continent (USA, Canada, and Chile), New Zealand, and the UK [7]. Peck and McCune [8] quantified bryophyte biomass and net moss accumulation across ten sites in northwestern Oregon, comparing areas within the Coast Range that had experienced intensive historical moss harvesting with those in the relatively undisturbed Cascade Range. Muir et al. [9] provided a comprehensive assessment of the volume and economic value of commercial moss harvest from forests in the Pacific Northwest and the Appalachian region, particularly West Virginia. To determine which taxa are most affected by the bryophyte harvest trade in West Virginia, Studlar and Peck [10] identified the mosses and liverworts acquired from commercial harvesters at a regional purchasing facility and at field sites deemed potentially harvestable. Gómez Peralta and Wolf [11] followed a group of about 10 moss gatherers in their harvesting activity and documented economic and ecological aspects. Over a single harvesting season, nearly 50 tons of fresh-weight bryophytes were removed from the forest floor, leaving behind a mosaic of gaps of bare soil in the moss layer in the forest of the Monarch Butterfly Biosphere Reserve, Sierra Chincua in Mexico. In a European context, bryophyte harvest is most evident in the UK [7]. Moss has been historically collected from the wild for the floral industry, principally funeral wreaths, and also for use as medical dressings (e.g., in World War I) [7]. Currently, fresh moss is mainly used for wreaths and floral arrangements [12] and in specialist horticulture (e.g., orchids) and is sold directly to the public in garden centers.Górriz-Mifsud et al. [7] examined commercial moss harvesting in Wales through the lens of the Socio-Ecological System (SES) framework, highlighting its role as a traditional small-scale forest activity contributing to rural livelihoods and income diversification. Moss picking, although declining, remains historically significant and is characterized by manual practices and limited technological input. The study situates this activity within the broader context of non-timber forest products (NTFPs), emphasizing the complex interactions between ecosystems, stakeholders, and governance structures.

At the European level, there is no explicit legislative prohibition on the commercial harvesting of mosses or bryophytes as a general activity. However, legal constraints arise indirectly through the conservation framework established under the EU Habitats Directive (formally known as Council Directive 92/43/EEC on the Conservation of Natural Habitats and of Wild Fauna and Flora). The study on moss picking in Wales highlights that most bryophytes are not individually listed as protected species under the Directive; nevertheless, they frequently occur within habitats designated under Annex I, thereby implicating them in habitat-level protection measures [7]. As such, harvesting activities that may lead to the degradation of these protected habitats are implicitly restricted or prohibited. From a broader European perspective, illegal bryophyte harvesting in forest stands operates within a fragmented and only partially harmonized regulatory framework. At the EU level, bryophytes are covered only indirectly by the Habitats Directive, which primarily protects habitats and selected species through the Natura 2000 network. Although the Habitats Directive provides the principal legal basis for conservation, its species-level protection of bryophytes remains highly selective. While Annex II identifies several rare bryophytes requiring the designation of Special Areas of Conservation, Annex V is limited to *Leucobryum glaucum* and all *Sphagnum* species except *S. pylaisii*, meaning that only a narrow subset of taxa is subject to potential exploitation control under Article 14. As a result, most bryophytes commonly harvested for decorative or commercial use fall outside explicit EU species protection and depend largely on habitat-based safeguards and national regulatory regimes. In Central Europe, *Leucobryum glaucum* ranks among the most frequently and deliberately collected moss species, consistently appearing at the top of regional bryological collection lists.

In Hungary, although collection in protected areas requires official permits and forest legislation restricts commercial extraction, illegal bryophyte harvesting has frequently been addressed primarily through property or forestry law rather than recognized as ecological degradation, weakening deterrence and conservation outcomes. Similar regulatory asymmetries are evident in other European countries, including Ireland and the United Kingdom, where Annex V taxa, particularly *Sphagnum* species, are subject to specific licensing systems or voluntary codes of practice, while the broader bryoflora remains largely unregulated at the species level. This limited taxonomic coverage under Annex V undermines coherent cross-border enforcement and shifts regulatory responsibility toward site-based protection mechanisms, thereby reinforcing the structural challenges in managing illegal bryophyte exploitation highlighted by this study.

The forested areas of Northeast Hungary are regularly affected by illegal moss harvesting, due to which official procedures have already been initiated in several cases. In previous cases, the procedures did not examine conservation issues, i.e., damage to species and habitats, but treated the matter as an act against property. In several cases, procedures were initiated regarding damage to natural values related to the collection of bryophytes, and these have already reached the court stage.

Bryophytes play fundamental roles in the functioning of forest ecosystems, influencing processes ranging from pedogenesis to microclimatic regulation and carbon cycling. Their contribution often extends beyond their small stature, as bryophyte carpets and cushions significantly affect ecosystem structure, hydrology, nutrient dynamics and biodiversity [13].

Bryophytes contribute to pedogenesis by enhancing rock weathering and facilitating the accumulation of organic matter. Through their decomposition, they release nutrients that become available to vascular plants and enrich the microbial communities associated with forest soils. By stabilizing fine sediments and creating microsites suitable for seedling establishment, bryophytes can strongly influence early successional dynamics [13].

Bryophyte mats can store substantial amounts of water relative to their biomass, thereby dampening fluctuations in moisture and temperature at the forest floor. In particular, *Sphagnum* species regulate hydrology by acidifying their environment, inhibiting decomposition and promoting peat accumulation, which makes peatlands globally important carbon sinks [13]. In non-peatland forests, pleurocarpous moss carpets also stabilize humidity, buffer evaporation, and maintain cool microhabitats essential for many soil invertebrates and seedlings.

Bryophytes support diverse communities of microorganisms, algae, fungi and invertebrates. Their fine-scale structural heterogeneity increases habitat availability and functional diversity within forest stands. As early colonizers in harsh environments, bryophytes can modify substrates, retain moisture and create favorable conditions for later-successional plant species, thus shaping long-term vegetation trajectories.

Bryophytes generally exhibit slow growth and recovery because their biomass accumulates gradually through apical or marginal extension. Growth rates vary widely among taxa and habitats, but many temperate forest mosses grow only 1–10 mm per year, and some species require several years to decades to rebuild fully developed carpets after disturbance [13].

Pleurocarpous moss carpets (including *Hypnum cupressiforme*) often regenerate over multiple years, depending on moisture and substrate stability. Cushion-forming acrocarps (e.g., *Dicranum*, *Grimmia*) can show faster shoot formation but require long periods to regain pre-disturbance biomass because individual cushions regrow slowly. In dry or nutrient-poor habitats, full bryophyte mat recovery may take 10–20 years, or longer if propagule sources are missing [13]. Experimental studies summarized by Glime indicate that even under favorable conditions, many forest bryophytes exhibit regeneration times of 3–5 years for partial cover and >10 years for complete mat restoration.

These regeneration dynamics underline the ecological sensitivity of bryophyte communities to harvesting and disturbance, particularly when large, monodominant carpets are removed.

In Hungary, according to the nature protection law, a permit is currently required for species collection in protected natural areas [14]. The Forest Law prohibits the commercial collection of bryophytes from tree bark in forest stands [15].

The main market for bryophyte materials is the floriculture industry, which uses *Hypnum cupressiforme* moss as a key component in moss wreaths in Hungary. The moss wreath is a popular decorative product, which is in high demand before 1 and 2 November (All Saints’ and All Souls’ Days in Hungary). These days, usually the Hungarian people visit the cemeteries and place some candles, candlesticks, flowers, and sometimes one or two moss wreaths on the surface of tombstones. The higher volume of illegal bryophyte collection takes place before November, usually in late summer and early autumn. Nevertheless, the moss wreath and other products are available in most of the decor and flower shops all year round and seasonally in hypermarkets in Hungary.

This paper presents the Hungarian legal framework, official procedures, and traditional practices related to bryophyte harvesting, alongside the taxonomic identification of illegally collected bryophyte material confiscated by the authorities in the summer of 2018 near the villages of Fony and Telkibánya (Figure 1 and Figure 2). In addition, the study compares the species composition of the seized material with that of the bryoflora in the affected forest stands, providing insight into the ecological implications of the indiscriminate harvesting activity.

## 2. Results

### 2.1. Moss Gatherers’ Practice in Hungary

Illegal moss collection is most prevalent in the hilly and mountainous regions of eastern Hungary. The main center of the uncontrolled moss trade operates in the Nyírség region, approximately 100 km from the affected forest stands, a distance that remains small enough to minimize the likelihood of roadside inspections and detection. According to observations by nature conservation guards, moss gatherers typically work in small groups within forest stands situated close to settlements. The collected bryophyte material is placed directly into second-hand polypropylene bags of various sizes, with gatherers manually compressing the contents to optimize space. More experienced harvesters attempt to remove entire moss carpets from rocks and tree bark in the largest pieces possible (Figure 3, Figure 4, Figure 5 and Figure 6), as intact, larger mats presumably hold greater market value than fragmented material.

Although gatherers are generally aware that their activity is illegal, many continue to collect seasonally due to the substantial financial incentive. Current identification results indicate that harvesters deliberately target *Hypnum cupressiforme*, with other species occurring only in low proportions within the collected material. Harvesting efforts concentrate on the bark of *Quercus* trees and on andesite rock surfaces, while terricolous bryophyte species and soil substrates are consistently absent from the seized samples.

### 2.2. Estimated Quantities and Species Composition of Bryophytes in the Confiscated Material

Altogether 16 bryophyte taxa (1 liverwort and 15 mosses) were identified from the harvested material. Based on the review and microscopic examination of the material confiscated in connection with the criminal offense (Figure 4 and Figure 5), it can be concluded that the moss *Hypnum cupressiforme* (8.7 m^3^) was collected illegally in the largest quantity. Additional species occurred only in negligible amounts in the bryophyte material. Most frequently *Paraleucobryum longifolium* (3 dm^3^) and *Dicranum scoparium* (2 dm^3^), and less frequently *Barbilophozia barbata* (2 cm^3^), *Brachytheciastrum velutinum* (1 cm^3^), *Brachythecium rutabulum* (2 cm^3^), *Hedwigia ciliata* (5 cm^3^), *Isothecium alopecuroides* (3 cm^3^), *Plagiomnium cuspidatum* (1 cm^3^), *Plagiomnium rostratum* (1 cm^3^), *Platygyrium repens* (0.5 cm^3^), *Pleurozium schreberi* (2 cm^3^), *Ptychostomum moravicum* (1 cm^3^), *Pylaisia polyantha* (1 cm^3^), *Syntrichia ruralis* (3 cm^3^) and *Syntrichia virescens* (1 cm^3^) were collected from the surface of andesite rocks and the bark of *Quercus petraea* (Table 1).

### 2.3. Comparison of Species Composition: Illegally Harvested Material vs. Affected Forest Stands

After describing the bryophyte flora of the affected forest stands, it became evident that the overall flora is considerably more diverse than the species composition represented in the illegally harvested material. Consequently, the latter does not adequately reflect the bryophyte assemblages characteristic of the area. Altogether, 46 bryophyte taxa (3 liverworts and 43 mosses) were recorded from the damaged forest habitats within the Telkibánya and Fony village boundaries (Table 2). Of these, 34.8% (16 species) were affected by illegal collection, whereas the majority of species (65.2%) were not present in the confiscated material.

In general, terricolous species (e.g., *Atrichum undulatum*, *Fissidens taxifolius*, *Dicranella heteromalla*, *Plagiomnium affine*, *Polytrichum formosum*), species forming individual tufts or cushions, as well as infrequent taxa (e.g., *Orthotrichum* spp., *Ulota crispa*, *Grimmia* spp.) were absent from the harvested bryophyte material.

A large cushion (approximately 1 m^2^) of *Antitrichia curtipendula* was also discovered in the affected habitat near Fony village; however, this species was not included in the confiscated bryophyte material. *Antitrichia curtipendula* is listed as Endangered (EN) on the Hungarian Red List of bryophytes [17].

## 3. Discussion

### 3.1. Discussion of the Results

This study provides the first detailed report of a large-scale bryophyte harvesting event in Europe. Based on the species composition of the confiscated material, it is evident that moss gatherers purposefully target *Hypnum cupressiforme*, while other bryophytes appear only as incidental bycatch in negligible amounts. The cosmopolitan *H. cupressiforme* is widely used as a raw material in the decoration and floriculture industries, owing to its broad distribution in the temperate zone and its characteristic thick, monodominant carpets. Bučar et al. [6] similarly reported *H. cupressiforme* as the most frequently collected bryophyte in Croatia, primarily harvested for Christmas decoration products. The popularity of *H. cupressiforme* is driven by market demand: the species provides an excellent and unique raw material in domestic floriculture and the creative hobby sector. In addition to its abundance and biomass, *H. cupressiforme* offers several practical advantages, including its tough, mat-like structure; flexibility; easy moldability (even when cut); persistent green color; and esthetically appealing natural appearance.

The comparison between the illegally harvested material and the bryophyte flora of the affected forest stands revealed that the local flora is substantially more diverse than what is represented in the confiscated biomass. Terricolous mosses, species forming discrete tufts and cushions, and generally infrequent bryophytes were completely absent from the harvested material. This selective pattern reflects the collectors’ focus on bryophytes growing on andesite rocks and the lower bark of *Quercus petraea*, where *H. cupressiforme* carpets are most abundant. Rare epiphytic taxa on higher trunk sections (e.g., *Orthotrichum* spp., *Ulota* spp., *Radula complanata*) and tuft-forming species on rocky substrates (e.g., *Grimmia* spp.) were therefore largely avoided or only accidentally removed. Time pressure and the risk of detection may further reinforce this selective behavior.

The bryophyte diversity of the affected forests was only moderately impacted; a considerable proportion of the local bryophyte flora does not appear in the harvested material. Most accompanying species were present only in minimal abundance. The collected bryophytes fall into the Hungarian species-frequency categories ‘Very Common’, ‘Common’ or ‘Widespread’ [16], and all belong to the Least Concern (LC) category of the Hungarian Bryophyte Red List, except *Barbilophozia barbata*, which is classified as Least Concern—attention category (LC-att) [16]. A large cushion (approx. 1 m^2^) of *Antitrichia curtipendula*—an Endangered (EN) species—was observed in the affected habitat, yet was absent from the confiscated biomass. However, it is important to emphasize that the increasingly common, uncontrolled harvest of large quantities of bryophytes poses a significant threat to bryophyte diversity at the affected sites. The long regeneration time of extensive bryophyte colonies means that repeated harvesting of this magnitude, especially at the same sites, can ultimately drive the decline of even common taxa.

Illegal harvesting has also recently expanded beyond areas near settlements, affecting additional protected areas within Aggtelek National Park. Such activities raise several nature conservation concerns, especially because bryophyte carpets function as integral habitat structures and not merely as aggregations of individual species. Their removal affects bryophyte vegetation-associated microfauna, water balance and local microclimatic conditions. In the last few years, we have initiated experiments on the temporal and spatial regeneration of bryophytes to assess recovery rates on stripped rock and bark surfaces. These studies aim to provide a scientific basis for future expert assessments.

In Hungary, the Act on the Protection of Nature (1996) regulates bryophyte collection in protected areas by requiring a permit [14]. In non-protected forest areas, the Forest Act (2009; § 61 (1) c) prohibits the commercial collection of mosses from the bark of living trees [15]. To date, official procedures have focused primarily on individual moss gatherers, while the market for bryophyte products remains largely unregulated. As a result, intensive bryophyte harvesting is likely to continue in the forested regions of northeastern Hungary.

### 3.2. Supply Chain Responsibility and Legal Implications

This case study highlights the broader issue of responsibility within the supply chain of harvested bryophytes. Illegally gathered moss is typically handled at the initial stages of trade—by collectors, intermediaries and early sellers—while processed products entering the floriculture and retail sectors often obscure their origin. Decorative moss products, such as wreath bases, rarely carry documentation on their provenance, making it nearly impossible for downstream buyers to verify legality. Consequently, legal responsibility is usually confined to the collectors and first-level buyers, who may be prosecuted for theft, receiving stolen goods or administrative violations related to unauthorized collection.

### 3.3. Burden of Proof and Traceability

The burden of proof regarding the legality of harvested bryophyte material lies with the authority initiating the procedure, typically the police or the nature conservation ranger service. Yet effective mechanisms ensuring traceability are largely absent. While, in theory, legal collection could be authorized in some regions, current illegal practices bypass all formal requirements—such as obtaining landowner or statutory permits—indicating that the problem does not stem from deficiencies in the regulatory framework itself, but rather from its circumvention in practice. The lack of documentation, identification protocols or chain-of-custody systems renders bryophyte material virtually untraceable once it enters the commercial chain.

### 3.4. Broader Ecological Consequences: Impacts on Ecosystem Services

Beyond the direct impacts on species composition, illegal bryophyte harvesting may influence a wide range of ecosystem services provided by bryophytes in forest ecosystems. Bryophytes are key contributors to pedogenesis, water regulation, nutrient cycling and habitat provisioning, and therefore their removal can have cascading ecological consequences well beyond the loss of bryophyte biomass itself. As highlighted by Glime [13], bryophyte carpets exert disproportionate influence on forest functioning relative to their size.

Their roles in soil formation, organic matter accumulation and sediment stabilization directly affect forest regeneration and soil development. Their decaying biomass releases nutrients that support vascular plant establishment and sustain diverse microbial communities. Bryophyte mats also buffer microclimatic extremes by storing water and reducing evaporation, creating cool and humid microsites essential for soil invertebrates, cryptogams and early plant stages. The removal of *H. cupressiforme* carpets can therefore alter near-ground microclimates and intensify drying processes, negatively impacting associated taxa.

Bryophytes additionally support diverse microbial and invertebrate communities through their fine-scale structural heterogeneity; harvesting eliminates these microhabitats, reducing habitat complexity and increasing ecosystem vulnerability. Importantly, the regeneration capacity of bryophyte communities is limited: many temperate forest mosses grow only 1–10 mm per year, and complete recovery of disturbed carpets often requires years to decades. Pleurocarpous species may take several years to regain partial cover, and more than a decade is often necessary for full mat reconstruction, particularly on exposed substrates or where propagule availability is low. Slow recovery means that repeated or extensive harvesting can result in long-term degradation with cumulative effects on ecosystem services.

Taken together, these ecological functions and slow recovery dynamics demonstrate that the consequences of illegal bryophyte harvesting extend far beyond the bryophyte layer itself. By altering soil formation, hydrology, microclimatic buffering and habitat provisioning, bryophyte removal can initiate cascading effects on multiple taxa and key ecosystem processes. Recognizing these wider ecological implications strengthens the argument that illegal moss harvesting represents a significant and underappreciated conservation concern in forest ecosystems.

### 3.5. Policy and Governance Recommendations

Several opportunities exist to improve regulatory oversight and conservation outcomes. First, documentation and monitoring systems could be strengthened through traceability requirements for commercial moss products and clearer due diligence obligations for buyers. Second, harmonized reporting and enforcement mechanisms among EU Member States would improve cross-border regulation of bryophyte trade, particularly as market demand and supply chains often extend beyond national boundaries. Third, the current EU legislation restricts bryophyte harvesting only within *Leucobryum* and *Sphagnum*. However, as this case demonstrates, harvested bryophyte carpets should be interpreted not only as collections of individual species but as integral habitat structures whose removal affects microclimate, biodiversity, nutrient cycling and ecosystem stability. Expanding regulatory attention from species-level protection to habitat-level considerations would therefore be advisable.

Finally, enhancing public and commercial awareness of supply chain responsibility could help reduce demand for illegally harvested bryophytes. Clearer guidelines for floriculture businesses and retailers regarding due diligence obligations, together with increased transparency in sourcing, could significantly contribute to limiting illegal moss harvesting in Central and Eastern Europe.

## 4. Materials and Methods

### 4.1. Study Area

The affected and surveyed localities, Fony and Telkibánya, are situated in the Abaúji Hills and Central Zemplén microregions, along the western margin of the Zemplén Mountains in northeastern Hungary (Figure 1). The bedrock is predominantly andesite, which frequently emerges at the surface within the forest stands (Figure 2). The study area lies in the colline zone at 130–540 m a.s.l. The climate is moderately cool and moderately dry, with a mean annual temperature of 8.0–9.5 °C. Annual precipitation ranges between 600 and 650 mm, of which 370–450 mm falls during the growing season. The most affected habitats are older acidophilous oak stands located near the settlements.

The forest stands of the two study areas are dominated by mature sessile oak (*Quercus petraea*) communities with varying admixtures of other broadleaved species. In Telkibánya, the stands form mixed sessile oak forests with *Carpinus betulus*, *Fagus sylvatica*, *Acer campestre*, *Prunus avium*, and *Pinus sylvestris*. Although not legally protected, the site belongs to a locally designated Natura 2000 area, with timber production as its primary land-use function. The soils are brown forest soils with acidic pH (4.2–5.2). In contrast, the Fony site is legally protected and primarily designated for nature conservation, with additional soil protection and Natura 2000 functions. The stands consist of mature sessile oak mixed with *Carpinus betulus*, *Quercus cerris*, *Acer campestre*, and *Prunus avium*. The area is also underlain by brown forest soils exhibiting acidic pH (4.5–5.5).

### 4.2. Harvested Bryophyte Material

The illegally harvested bryophyte material was seized by Hungarian authorities in August 2018 near Fony and Telkibánya (Figure 1). Approximately 8.7 m^3^ (air-dry weight: 296 kg) of collected bryophytes was sorted by the first author at the Encs Police Station (Figure 4). During the on-site sorting, selected samples were set aside for microscopic identification. After species-level identification of the bryophyte material confiscated by the authorities, volumetric quantification was performed using graduated cylinders of appropriate volume. For *Hypnum cupressiforme*, which constituted the largest proportion of the material, volume measurements were conducted using calibrated large polypropylene bags with known capacity.

Bryological surveys of the impacted forest stands were carried out in August, October, and December 2021 (GPS coordinates: 48.406757° N, 21.287531° E; 48.394860° N, 21.306072° E; 48.498942° N, 21.384923° E).

Nomenclature follows Hodgetts et al. [18] for liverworts and mosses. Voucher specimens are deposited in the Cryptogamic Herbarium of the Department of Botany and Plant Physiology, Eszterházy Károly Catholic University, Eger (EGR). Some contextual information derives from reports provided by national park conservation rangers.

## Figures and Tables

**Figure 1 plants-14-03785-f001:**
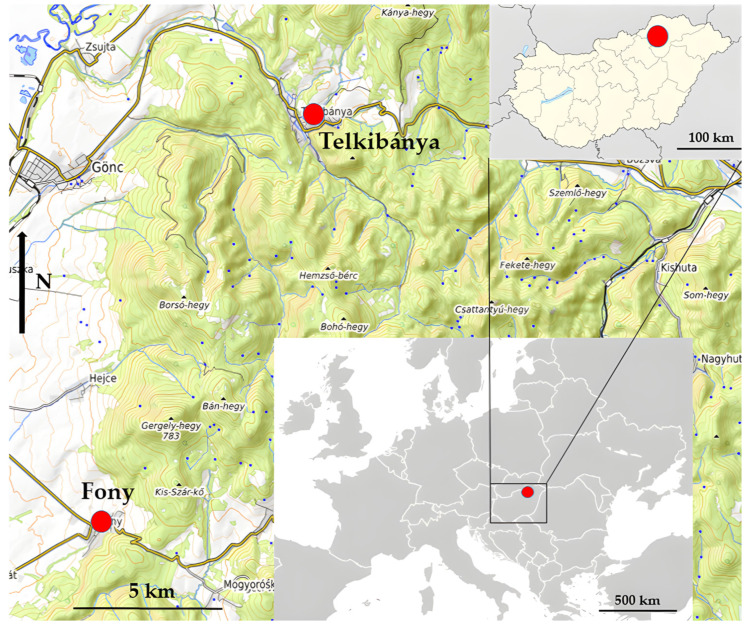
Locations affected by illegal bryophyte harvesting in the Zemplén Mountains (NE Hungary). Source: © OpenStreetMap contributors and wiki-vr; map prepared by Péter Szűcs.

**Figure 2 plants-14-03785-f002:**
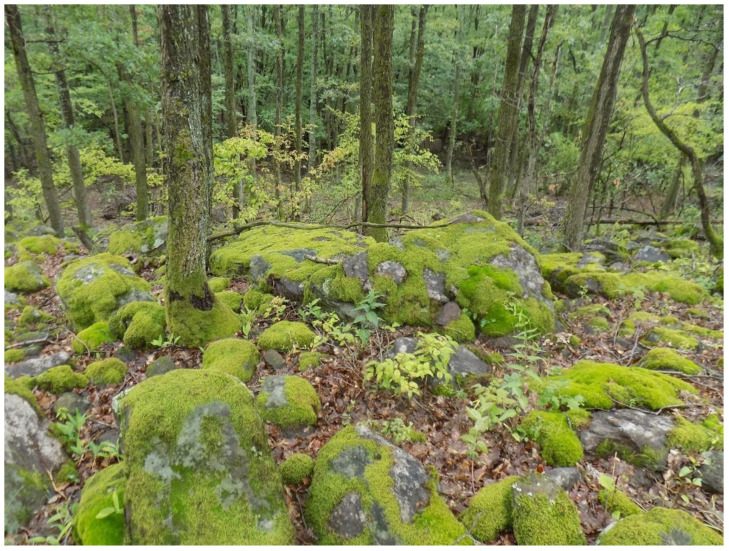
Habitat of *Paraleucobryum longifolium*, *Dicranum scoparium*, *Barbilophozia barbata*, and *Hypnum cupressiforme* on andesite rock surfaces within a degraded *Quercus* forest stand near the village of Fony.

**Figure 3 plants-14-03785-f003:**
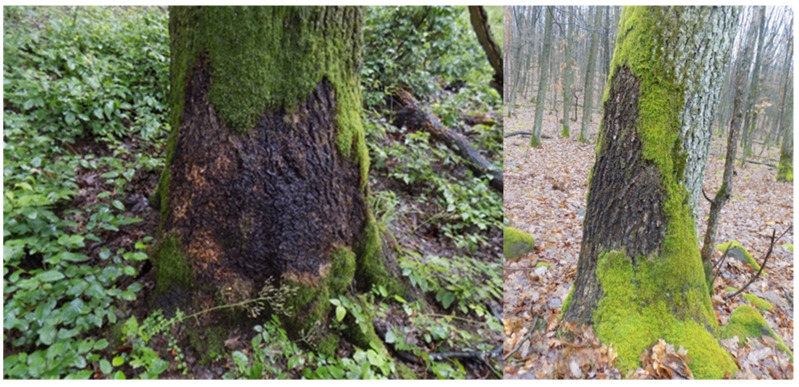
Removed *Hypnum cupressiforme* carpets on the bark of *Quercus* trees in a forest stand near the village of Telkibánya.

**Figure 4 plants-14-03785-f004:**
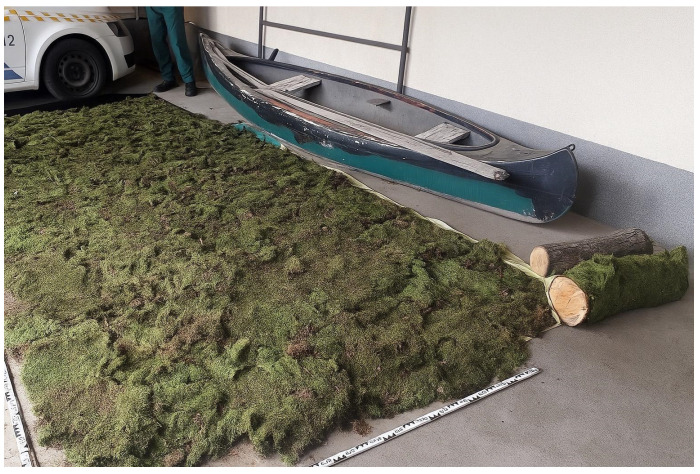
Part of the confiscated bryophyte material was used as evidence at the police station in the town of Encs.

**Figure 5 plants-14-03785-f005:**
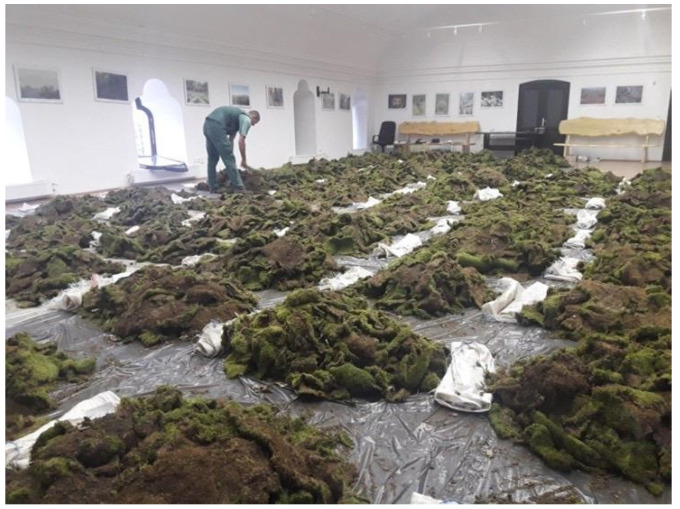
Drying process of the illegally harvested and confiscated bryophyte material in the building of Aggtelek National Park, Bodrogkeresztúr.

**Figure 6 plants-14-03785-f006:**
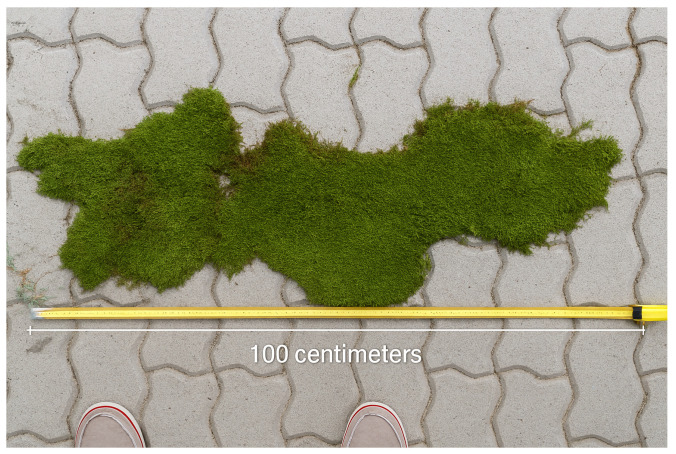
A 100 cm-long, intact *Hypnum cupressiforme* carpet from the harvested material.

**Table 1 plants-14-03785-t001:** Frequency, national Red List status, and estimated quantities of bryophyte species identified from illegally harvested material. Samples originated from the bark of *Quercus petraea* and from andesite rock surfaces. Species are presented in descending order of estimated quantity. LC: Least Concern; LC-att: Least Concern, attention category.

Bryophyte Species (Illegally Harvested)	Frequency in Hungary [16]	Red List Status in Hungary [17]	Estimated Quantity (cm^3^)
*Hypnum cupressiforme*	very common	LC	<8,700,000
*Paraleucobryum longifolium*	widespread	LC	3000
*Dicranum scoparium*	common	LC	2000
*Hedwigia ciliata*	widespread	LC	5
*Isothecium alopecuroides*	common	LC	3
*Syntrichia ruralis*	very common	LC	3
*Brachythecium rutabulum*	very common	LC	2
*Pleurozium schreberi*	widespread	LC	2
*Barbilophozia barbata*	widespread	LC-att	2
*Brachytheciastrum velutinum*	very common	LC	1
*Plagiomnium cuspidatum*	common	LC	1
*Plagiomnium rostratum*	widespread	LC	1
*Ptychostomum moravicum*	very common	LC	1
*Pylaisia polyantha*	very common	LC	1
*Syntrichia virescens*	common	LC-att	1
*Platygyrium repens*	very common	LC	0.5

**Table 2 plants-14-03785-t002:** List of bryophyte species from habitats affected by illegal harvest (Telkibánya and Fony sites). Species occurrence, their substrate, and Red List status are included. Bryophytes that were also found in the illegally harvested material are marked in bold. Species occurrence at the sites is marked with ‘+’. The bryophyte material originated from the bark of *Quercus petraea* and, in one case, *Fagus sylvatica*, as well as from andesite rock surfaces and soil. Threat categories: EN—Endangered; LC—Least Concern; LC-att—Least Concern, attention category; NT—Near threatened.

Bryophyte Flora of the Affected Habitats (Bryoflora of Illegally Harvested and Confiscated Material in Bold)	Telkibánya Site	Fony Site	Substrate	Red List Status in Hungary [17]
* Liverworts *	occurrence at the site			
***Barbilophozia barbata*** (Schmidel ex Schreb.) Loeske		+	andesite rock	LC-att
*Frullania dilatata* (L.) Dumort.	+	+	oak bark	LC
*Radula complanata* (L.) Dumort.	+	+	oak bark	LC
* Mosses *				
*Amblystegium serpens* (Hedw.) Schimp.	+	+	soil, bark	LC
*Anomodon viticulosus* (Hedw.) Hook. & Taylor		+	andesite rock	LC
*Antitrichia curtipendula* (Hedw.) Brid.		+	andesite rock	EN
*Atrichum undulatum* (Hedw.) P.Beauv	+	+	soil	LC
***Brachytheciastrum velutinum*** (Hedw.) Ignatov & Huttunen	+	+	soil, bark	LC
***Brachythecium rutabulum*** (Hedw.) Schimp.	+	+	soil, bark	LC
*Brachythecium salebrosum* (Hoffm. ex F.Weber & D.Mohr) Schimp.	+		bark	LC
*Climacium dendroides* (Hedw.) F.Weber & D.Mohr		+	soil	LC-att
*Dicranella heteromalla* (Hedw.) Schimp.	+	+	soil	LC
*Dicranum montanum* Hedw.	+	+	oak bark	LC
***Dicranum scoparium*** Hedw.		+	andesite rock	LC
*Fissidens taxifolius* Hedw.	+		soil	LC
*Grimmia hartmanii* Schimp.		+	andesite rock	LC
*Grimmia pulvinata* (Hedw.) Sm.		+	andesite rock	LC
***Hedwigia ciliata*** (Hedw.) P.Beauv.		+	andesite rock	LC
*Homalia trichomanoides* (Hedw.) Brid.		+	oak bark	LC-att
*Hylocomiadelphus triquetrus* (Hedw.) Ochyra & Stebel.		+	andesite rock	LC
***Hypnum cupressiforme*** Hedw.	+	+	soil, rock, bark	LC
***Isothecium alopecuroides*** (Lam. ex Dubois) Isov.		+	soil, bark	LC
*Leskea polycarpa* Hedw.	+		bark	LC
*Lewinskya affinis* (Schrad. ex Brid.) F.Lara, Garilleti & Goffinet	+	+	bark	LC
*Lewinskya speciosa* (Nees) F.Lara, Garilleti & Goffinet	+	+	bark	LC
*Orthotrichum pallens* Bruch ex Brid.	+		bark	LC
*Orthotrichum stramineum* Hornsch. ex Brid.		+	bark	LC
***Paraleucobryum longifolium*** (Hedw.) Loeske		+	andesite rock	LC
*Plagiomnium affine* (Blandow ex Funck) T.J.Kop.	+		soil	LC
***Plagiomnium cuspidatum*** (Hedw.) T.J.Kop.	+	+	soil	LC
***Plagiomnium rostratum*** (Schrad.) T.J.Kop.	+	+	soil	LC
***Platygyrium repens*** (Brid.) Schimp.	+	+	bark	LC
***Pleurozium schreberi*** (Willd. ex Brid.) Mitt.		+	andesite rock	LC
*Polytrichum formosum* Hedw.	+	+	soil	LC
*Pseudanomodon attenuatus* (Hedw.) Ignatov & Fedosov		+	bark	LC
*Pterigynandrum filiforme* Hedw.	+		beech bark	LC
***Ptychostomum moravicum*** (Podp.) Ros & Mazimpaka	+	+	oak bark	LC
*Pulvigera lyellii* (Hook. & Taylor) Plášek, Sawicki & Ochyra	+	+	oak bark	LC
***Pylaisia polyantha*** (Hedw.) Schimp.	+	+	oak bark	LC
***Syntrichia ruralis*** (Hedw.) F.Weber & D.Mohr		+	andesite rock	LC
*Syntrichia papillosa* (Wilson) Jur.	+	+	oak bark	LC-att
***Syntrichia virescens*** (De Not.) Ochyra	+	+	oak bark	LC-att
*Thuidium recognitum* (Hedw.) Lindb.		+	andesite rock	LC-att
*Tortula subulata* Hedw.	+		soil	LC
*Ulota crispa (Hedw.) Brid*		+	oak bark	NT
*Weissia controversa* Hedw.	+		soil	LC

## Data Availability

The original contributions presented in this study are included in the article. Further inquiries can be directed to the corresponding author. Collected bryophyte specimens are deposited and available at the Cryptogamic Herbarium of the Department of Botany and Plant Physiology at the Eszterházy Károly Catholic University, Eger (EGR).

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
