# Peer review of "Large-Scale Illegal Bryophyte Harvesting in Protected Areas of East-Central Europe, Hungary: Conservation Implications"

_plants, 2025, doi:10.3390/plants14243785_

Round 1

Reviewer 1 Report

Comments and Suggestions for Authors

The topic is very interesting, and the work provides a significant contribution to our understanding of the problem of illegal bryophyte collection in Europe. Furthermore, floristic data for two forest sites in Hungary are provided for the first time.
However, some conservation aspects are not adequately highlighted.
Therefore, some additions are needed to strengthen the text, as indicated in the attached file.

Reviewer 2 Report

Comments and Suggestions for Authors

Summary: Péter Szűcs et al. targeted illegal bryophyte harvesting in Europe, illustrated through a concrete case from a protected area in Hungary.

In general: The authors address a highly policy-relevant and underexplored issue: the illegal harvesting of mosses. For the first time, quantities are provided based on a concrete case from Hungary. Following linguistic improvements to enhance clarity, as well as some targeted additions to the policy context, publication of this study would be highly valuable and is, in my view, strongly recommended.
In addition, I believe that the positioning of the manuscript could be strengthened even further. Although the study is based on a specific case, the issue itself is of much broader relevance for European conservation policy. Emphasizing this wider context more clearly—while using the Hungarian case to illustrate and substantiate the problem with concrete data—would help ensure that the manuscript is perceived not only as a case study, but as a contribution to a significant and emerging policy challenge.

Title: The current title is somewhat lengthy and could be made more concise to enhance readability and clarity. In addition, I would encourage the authors to consider whether the title should signal more strongly that the manuscript addresses a broader and timely conservation problem, which is illustrated through the Hungarian case study rather than limited to it. Given the growing relevance of illegal bryophyte harvesting in Europe, a title that highlights the wider issue—supported by the case study—might increase the visibility and perceived importance of the work.

A possible alternative formulation could be: „Large-Scale Illegal Bryophyte Harvesting in Protected Areas of East-Central Hungary: Conservation Implications“ or potentially an even more general version that foregrounds the broader problem before specifying the case study.

Abstract: I suggest adding one or two sentences at the end of the abstract to better situate the study within the broader EU policy context. Given the relevance of illegal bryophyte harvesting for biodiversity conservation and enforcement, it would strengthen the abstract to briefly mention how the findings relate to existing EU regulations and to outline a concrete implication for management or policy action. Even a concise recommendation—such as the need for improved monitoring, stricter enforcement, or harmonized reporting across Member States—would substantially enhance the abstract’s impact and relevance.

Introduction: The manuscript provides valuable information on moss harvesting in the USA. Are there any additional documented cases from other EU Member States—perhaps in the form of grey literature, reports from conservation agencies, or media investigations—that could help situate this case within a broader European context. Even a brief mention of further examples, if available, could strengthen the argument that the issue extends beyond a single national case and thereby increase the overall weight and relevance of the study.

In addition, it might be helpful to include a short overview of the legislative framework in Europe. For instance, the authors could briefly mention relevant aspects of the EU Habitats Directive (e.g., Annex V for the genera Leucobryum and Sphagnum) and, if information is available, highlight how regulations in other European countries compare with the Hungarian situation already described. This could further contextualize the conservation and enforcement challenges addressed by the study.

Comment on Section 2.1: The paragraph describing official procedures, legal regulations in Hungary, and the market demand around All Saints’ Day provides useful contextual information. However, my impression is that this content functions primarily as background rather than as a result of the study. Would it perhaps fit better in the Introduction, where it could help frame the problem, contextualize the drivers of illegal bryophyte harvesting (e.g., market demand), and outline the national regulatory setting before the empirical results are presented?

Relocating this material to the Introduction might improve the flow of the manuscript by keeping Section 2 focused on the actual results, while simultaneously strengthening the background section with relevant socio-economic and legal context.

Comment on Table 2: In the heading of Table 2, “are also found” should be corrected to “were also found.”

Line 113, Figure 5. „Drying progress of the illegally harvested and confiscated bryophyte material“ – I think it should be „drying process“

Line 193: The phrase “very common, common, and widespread” appears to contain a redundant use of “common.” Please rephrase.

Comment on the discussion: The Discussion provides a helpful overview of the legislative context in Hungary. It may further strengthen the manuscript to place these findings within the broader European policy framework. For instance, the authors could briefly reflect on the fact that only a limited number of bryophyte genera are listed under Annex V of the EU Habitats Directive (Leucobryum, Sphagnum), and what implications this has for enforcement and cross-border regulation.

In addition, the manuscript might benefit from a short discussion of responsibilities along the supply chain. For example: To what extent can floriculture businesses or retailers be held accountable for purchasing and selling bryophytes of potentially illegal origin? Where does the burden of proof lie regarding the legality of collected material, and are there mechanisms in place (or lacking) to ensure traceability?

Finally, it could be valuable to outline possible policy or management recommendations emerging from the case study—such as improving documentation and monitoring systems, strengthening the obligations of commercial buyers, or encouraging harmonized reporting and enforcement across EU Member States. At the moment collection of bryophytes is regulated only for the genera Leucobryum and Sphagnum. Even a concise reflection on these points would considerably enhance the policy relevance of the Discussion.

Further comment on the Discussion: I would also suggest expanding the section on the ecosystem services provided by bryophytes in forest ecosystems (line 201). Drawing more explicitly on the relevant literature could help illustrate that bryophyte harvesting does not only affect the bryophyte communities themselves, but may also have cascading consequences for other taxa and for key ecosystem services. Highlighting these broader ecological implications would strengthen the argument for why illegal bryophyte harvesting represents a significant conservation concern.

Comments on the Quality of English Language

Although I am not a native speaker myself, my impression is that the text is somewhat wordy and includes several expressions that are unusual. At times, the flow of information can be difficult to follow.
You might consider streamlining the text by simplifying overly long sentences, reducing repeated phrasing, and replacing certain non-idiomatic expressions with more widely used scientific terminology (e.g., “legal regulations” instead of “law regulations,” “taxonomic identification” instead of “taxonomical identification,” etc.).

Round 2

Reviewer 2 Report

Comments and Suggestions for Authors

Congratulations for your efforts, the revised version is clearly improved.

Comments on the Quality of English Language

Although I am not a native speaker myself, my impression is that the text is somewhat wordy and includes several expressions that are unusual. At times, the flow of information can be difficult to follow.
You might consider streamlining the text by simplifying overly long sentences, reducing repeated phrasing, and replacing certain non-idiomatic expressions with more widely used scientific terminology (e.g., “legal regulations” instead of “law regulations,” “taxonomic identification” instead of “taxonomical identification,” etc.).